# Population-Based Analysis of the Immunoglobulin G Response to Different COVID-19 Vaccines in Brazil

**DOI:** 10.3390/vaccines11010021

**Published:** 2022-12-22

**Authors:** Nigella M. Paula, Marcelo S. Conzentino, Ana C. A. Gonçalves, Renata da Silva, Karin V. Weissheimer, Carlos H. S. Kluge, Paulo H. S. A. Marins, Haxley S. C. Camargo, Lucas R. P. Farias, Thamyres P. Sant’Ana, Letícia R. Vargas, Juliane D. Aldrighi, Ênio S. Lima, Guiomar T. Jacotenski, Fabio O. Pedrosa, Alan G. Gonçalves, Emerson Joucoski, Luciano F. Huergo

**Affiliations:** 1Setor Litoral, Federal University of Paraná, UFPR, Matinhos 83260-000, PR, Brazil; 2Graduated Program in Sciences–Biochemistry, UFPR, Curitiba 81531-980, PR, Brazil; 3Graduated Program in Farmacy, UFPR, Curitiba 80060-240, PR, Brazil

**Keywords:** SARS-CoV-2, humoral response, vaccine

## Abstract

(1) Background: COVID-19 vaccination in Brazil has been performed mostly with CoronaVac (Sinovac), ChAdOx1-S (AstraZeneca-University of Oxford) and BNT162b2 (Pfizer-BioNTech) vaccines. The titers of IgG antibodies reactive to the SARS-CoV-2 spike protein correlate with vaccine efficacy. Studies comparing vaccine immunogenicity in a real-world scenario are lacking. (2) Methods: We performed a population-based study to analyze the immunoglobulin G response to different COVID-19 vaccines. Citizens older than 18 years (n = 2376) provided personal data, a self-declaration of any previous COVID-19 positive tests and information regarding COVID-19 vaccination: the vaccine popular name and the date of each dose. Blood samples were collected and the levels of IgG reactive to SARS-CoV-2 antigens were determined and compared between different vaccine groups. (3) Results: The seroconversion for anti-spike IgG achieved > 95% by February 2022 and maintained stable until June 2022. Higher anti-spike IgG titers were detected in individuals vaccinated with BNT162b2, followed by ChAdOx1-S and CoronaVac. The anti-spike IgG response was negatively correlated with age and interval after the second dose for the BNT162b2 vaccine. Natural infections boosted anti-spike IgG in those individuals who completed primary vaccination with ChAdOx1-S and CoronaVac, but not with BNT162b2. The levels of anti-spike IgG increased with the number of vaccine doses administered. The application of BNT162b2 as a 3rd booster dose resulted in high anti-spike IgG antibody titers, despite the type of vaccine used during primary vaccination. (4) Conclusions: Our data confirmed the effectiveness of the Brazilian vaccination program. Of the vaccines used in Brazil, BNT162b2 performed better to elicit anti-spike protein IgG after primary vaccination and as a booster dose and thus should be recommended as a booster whenever available. A continuous COVID-19 vaccination program will be required to sustain anti-spike IgG antibodies in the population.

## 1. Introduction

The high transmissibility of SARS-CoV-2 along with the rapid growing COVID-19 fatalities has led to massive global efforts to develop effective vaccines. The main immunogenic components of SARS-CoV-2 are the nucleocapsid and spike proteins. Strong humoral IgG response against these antigens is detectable in COVID-19 convalescent cases [1]. The nucleocapsid protein is located within the viral particle in close contact with the viral RNA. The spike protein is surface located and mediates the binding of SARS-CoV-2 to the angiotensin converting enzyme 2 receptor (ACE2) present on the host cell membrane through the spike S1 receptor-binding-domain (RBD) [2]. Therefore, most of the COVID-19 vaccines in use or under development were designed to induce an immune response to the spike protein antigen [3].

In Brazil, the COVID-19 vaccination program is using CoronaVac (Sinovac), which is based on inactive SARS-CoV-2; BNT162b2 (Pfizer-BioNTech), which is based on mRNA technology; and ChAdOx1-S (AstraZeneca-University of Oxford) and Ad26.COV2.S (Janssen), which are based on adenoviral vectors. The vaccines present only the full-length spike protein to the immune system, the exception being CoronaVac, which is based on the whole virus.

Clinical studies involving the vaccines in use in Brazil reported efficacy against symptomatic infection after the 2nd shot of 95% for BNT162b2 [4], 62–67% for ChAdOx1-S [5], 50–84% for CoronaVac [6] https://www.bbc.com/news/world-latin-america-55642648 [7] (accessed on 1 May 2022) and 70% for Ad26.COV2.S (after one shot) [8]. Despite these numbers, vaccine efficacy may be different in a real-world scenario as it may be affected by other variables, including challenging logistics, delays on second shot due to vaccine shortages and/or public hesitancy, among others.

One of the major mechanisms of protection elicited by vaccines involves the production of IgG antibodies reactive to spike protein and to S1 RBD, the levels of which are well correlated to each other as well as with virus neutralization activity. The levels of anti-spike and anti-RBD IgG are negatively correlated with the risk of symptomatic COVID-19 and hence can be used to reliably predict the level of vaccine efficacy [9,10].

Some studies have reported that COVID-19 vaccines deployed in Brazil can elicit IgG reactive to the spike protein antigen [11,12,13,14]. However, these studies were not designed to compare IgG levels obtained using different vaccines, they were limited to a small cohort and/or a specific group (i.e., heath care professionals) and/or did not provide information regarding the duration of the IgG response, the effectiveness of multiple vaccine doses/boosters nor to the effects of natural infections.

Here we performed a population-based analysis of the immunoglobulin G response to different COVID-19 vaccines available in Brazil.

## 2. Materials and Methods

### 2.1. Study Design and Sampling

To understand the humoral response to SARS-CoV-2 in a real-world scenario, no specific public groups were enrolled for this study and anyone older than 18 years was invited to participate. Participants could participate at any time for the sampling campaigns. Members of the public were invited through advertisements on the internet, radio and television. Sampling campaigns were performed weekly between January 2021 and June 2022 at the campus of the Federal University of Paraná in the city of Matinhos.

Participants were directed to the study web site http://200.17.236.32/covid19/ where they could choose from any available date and time to come to the study center for sampling. A questionnaire was presented online to collect personal information including age, sex and the city of residence; a self-declaration of the dates of any previous COVID-19 positive tests, a self-declaration of previous COVID-19 vaccine, the vaccine manufacture’s popular name (AstraZeneca, CoronaVac, Janssen or Pfizer) and the date of the first, second and third doses (if any). All information was stored in a MySQL database. Informed consent declaration was obtained from all participants.

### 2.2. Serological Analysis

Blood was collected using capillary puncture, the serum was used to investigate IgG reactive against three different SARS-CoV-2 antigens, nucleocapsid (anti-N), spike (anti-S) and S1 RBD (anti-RBD). Briefly, 1 mg of His-tagged antigens purified as described previously [15,16,17,18,19] were incubated with 1 mL of nickel magnetic particles (Promega—V8565) in 50 mL of TBST. After 5 min at room temperature with gentle mix, the beads were washed with 25 mL of TBST and resuspended in 5 mL of TBST. The loaded beads were stored in 0.8 mL aliquots at 4 °C.

The magnetic bead immunoassay was performed using the 96-sample format with flat bottom plates (Cralplast). The 0.8 mL aliquots of antigen loaded beads were resuspended in 11 mL of TBST containing 1% (*w/v*) skimmed milk and 0.1 mL of the mixture was distributed in each well of a 96-well plate. Four micro liters of human serum were diluted in 0.2 mL of TBST 1× skimmed milk 1% (*w/v*) directly on the wells of a second 96-well plate. The magnetic beads were transferred to the sample plate and incubated with the human sample (serum or blood) for 2 min with gentle mix. The beads were captured and loaded into sequential 2-wash steps for 30 sec in 1x TBST. The beads were incubated for 2 min with 0.15 mL goat anti-human IgG-PE (Moss Inc., Franklin Park, IL, USA) diluted 1:250 in 1× TBST, followed by a 2-wash step for 30 sec in 1× TBST. The beads were transferred to a final plate containing 0.15 mL of TBST in each well and homogenized for 10 s followed by fluorescent reading using a TECAN M Nano plate reader (TECAN) operating at fluorescent top reading. Excitation 545 nm (bandwidth 9 nm and 25 flashes) and emission 578 nm (bandwidth 20 nm, integration time 20 µs and Z-position at 20,000 µm).

The presence of reactive IgG to anti-N and anti-S in the samples were investigated using a magnetic immunofluorescence assay operating at >99.5%, specificity and >95% sensitivity as described previously [18]. The presence of high avidity IgG reactive to anti-S1 RBD were performed in chromogenic format under stringent conditions using 1 mol·L^−1^ of urea in the wash buffers as described previously operating at a specificity and sensitivity of >82% and >98%, respectively [15]. Raw fluorescent or absorbance values in each sample were normalized using a reference serum and expressed as a % of the reference. Seroconversion rates were defined as the % of IgG positive tests in relation to the total number of individuals analyzed.

### 2.3. Statistical Analysis

Statistical analyses were performed using NCSS 11, GraphPad Prism 8 and RStudio R 3.6.1 using the packages “FactoMineR” and “factoextra”. Multiple comparisons of IgG levels were performed by applying a one-way ANOVA Tukey test. Adjusted two-tailed *p* values are reported. Populational data confidence intervals were calculated considering positive prevalence of 10% for nucleocapsid (N) and 50% for spike protein (S) and spike S1 RBD (RBD), with reference to the population size of 35,705 for the city of Matinhos. Seroprevalence results were compared with official numbers of COVID-19 cases and deaths in the city of Matinhos provided by the health authorities.

## 3. Results

Weekly sampling campaigns were open to the public between January 2021 and June 2022 in the city of Matinhos, in the coastal region of Paraná state, in the south of Brazil (Appendix A). A total of 2376 samples were collected from 1785 different individuals. All samples were analyzed for the presence of anti-N IgG. Samples collected since July 2021 were also analyzed for anti-S IgG (n = 1980). In addition, 1225 samples were evaluated for the presence of high avidity IgG antibodies reactive to S1 RBD. In total, 5581 analyses were performed. Although the study site was in the city of Matinhos, visitors and workers residing in other cities were also enrolled. The cohort was composed of residents of Matinhos (60%), Curitiba (26%; capital of the Paraná state), Paranaguá (6.4%), Guaratuba (3.5%) and from other cities (2.8%) (Appendix A). The age of the participants had a mean of 40 years (SD 13; min. 18 max. 86 years old) with a 65% predominance of women.

### 3.1. Seroconversion Rates for Nucleocapsid, Spike Protein and S1 RBD

To understand the effectiveness of the vaccination program in Brazil, it is important to estimate the fraction of the population that had experienced natural infection. These numbers can be determined based on the fraction of those individuals with a positive IgG test for the N antigen after excluding individuals vaccinated with CoronaVac, the only vaccine applied in Brazil that can elicit anti-N antibodies. In our previous serological, 16.7% (12.6–20.7, 95% CI) of the population had experienced COVID-19 by the end of 2021 [14]. This number raised sharply in 2022 after the introduction of the SARS-CoV-2 omicron variant in region, reaching 26.1% (23.3–28.5, 95% CI) in the first trimester of 2022. This number remained stable in the second trimester of 2022, 25.6% (21.4–29.8, 95% CI) (Appendix A). The data indicate that more than a quarter of the cohort had experienced SARS-CoV-2 infection by March 2022. It is worth mentioning that the real number of cases should be higher as IgG anti-N sero-revertants had been detected in this cohort during our previous study [14].

We showed in our previous study that anti-spike IgG seroconversion in 2021 followed the trend of the population fraction that had taken the second dose [14]. The numbers in 2022 confirmed this trend. By February 2022, official numbers indicated that 96.4% of the eligible population had completed the primary vaccination (2nd dose). Accordingly, 95.6% (88.3–100%, 95% CI) of spike protein seroconversion was detected and spike protein seroconversion remained above 95% between February and June 2022 (Appendix A). Seroconversion for high avidity IgG reactive to SARS-CoV-2 S1 RBD antigen was 53.8% (48.6–58.9, 95% CI) in November 2021 reaching 70.4% (62.2–78.6, 95% CI) in February 2022. The numbers remained stable between 65 and 70% from February to June 2022 (Appendix A).

The data described above support the effectiveness of the Brazilian vaccination program concerning spike protein seroconversion. However, it is important to stress that a significant fraction of spike protein seroconverts could result from natural infections, estimated to have occurred in more than a quarter of the cohort during the first trimester 2022.

### 3.2. IgG Reactive to Spike Protein and S1 RBD upon Completion of Primary Vaccination

To depict the IgG response raised after completion of the primary vaccination (2nd dose) using vaccines from different manufactures, the cohort was stratified following the information provided by the participants. From this point on, unless stated otherwise, the following filters were applied to the cohort. Participants with self-declared previous COVID-19 positive tests were excluded to minimize inputs resulting from natural infections. Only participants who declared to have taken the second dose of the vaccine in the time frame between 10 and 240 days before sampling were considered. The remaining cohort consisted of 804 samples from 716 individuals distributed accordingly to the vaccine type as ChAdOx1-S (AstraZeneca) 45.4%, BNT162b2 (Pfizer) 32.7%, CoronaVac (Sinovac) 21.6% and Ad26.COV2.S (Janssen) 0.2%. Given the low representative, Ad26.COV2.S was excluded from the comparative analyses.

The IgG seroconversion rates in participants vaccinated with BNT162b2 were 99% and 77% for spike protein and S1 RBD, respectively. ChAdOx1-S positive rates were 81% for spike protein and 36% for S1 RBD. Volunteers vaccinated with CoronaVac were 52% and 31% positive for spike protein and S1 RBD, respectively. This trend in the seroconversion ratio was evident when the signal of IgG reactive to spike protein and S1 RBD were plotted accordingly to the vaccine type. The IgG levels were higher for BNT162b2, followed by ChAdOx1-S and CoronaVac. The differences were significant (*p* < 0.0001) in all comparisons except for ChAdOx1-S vs. CoronaVac in the case of IgG reactive to S1 RBD (*p* = 0.46) (Figure 1).

It is important to note that the average IgG levels for spike protein and S1 RBD were significantly higher (*p* < 0.0001) after completion of the primary vaccination using vaccines from different manufactures when compared to pre-pandemic samples (Figure 1). The only exception was CoronaVac, where IgG reactive to S1 RBD was not significantly different from pre-pandemic samples (Figure 1). These data confirm that all vaccines in use in Brazil were effective in activating the IgG response to the spike protein antigen. The levels of IgG reactive to spike protein and RBD were not influenced by sex in different vaccine types (Appendix A).

### 3.3. IgG Response According to Age and Interval after the Second Dose

To obtain insights into the IgG response among different age groups and over time, Pearson-correlation analyses were performed by plotting the anti-spike or anti-RBD IgG signal vs. age or time interval after the 2nd dose (primary vaccination). For the CoronaVac and ChAdOx1-S vaccines, there was no significant correlation between IgG levels and Age or time interval (Figure 2 and Appendix A). In contrast, for BNT162b2, significant (*p* < 0.0001) weak-to-moderate negative correlations were observed in those comparisons (Figure 2 and Appendix A). Hence, even though BNT162b2 performed better than CoronaVac and ChAdOx1-S to induce IgG response against spike and RBD, the ability of BNT162b2 to sustain such a high response was negatively influenced by age and the time after the 2nd dose.

The distribution of age and interval were not even within the *x* axis among different vaccine types (Figure 2). Furthermore, in the CoronaVac subgroup, there was a significant positive correlation between age and time after the 2nd dose (Figure 2). This is explained by the fact that CoronaVac was the first vaccine introduced in Brazil and mainly applied to the elderly. To compare vaccine immunogenicity without age and/or interval bias, the cohort was further filtered to those individuals of 18–40 years that received the 2nd dose 10–90 days before sampling. When the levels of IgG reactive to spike and RBD in this subgroup were plotted accordingly to the vaccine type, the same trend and statistics presented in Figure 1 was detected (Appendix A). These data confirm the higher immunogenicity of BNT162b2 followed by ChAdOx1-S and CoronaVac.

### 3.4. Comparison of Primary Vaccination Concurring with Natural Infection Accordingly to Vaccine Type

Individuals that completed the primary vaccination (2nd dose) were distributed according to vaccine type and separated into infected and non-infected subgroups (Figure 3A). For this analysis, IgG reactive to RBD was not considered due to the low number of samples in some subgroups. Participants who declared a previous positive diagnostic for COVID-19 were considered as the infected group despite the infections that occurred before or after vaccination.

Natural infections increased anti-spike IgG in the CoronaVac (*p* = 0.0173) and ChAdOx1-S (*p* < 0.0001) subgroups, but not in those individuals vaccinated with BNT162b2 (*p* = 0.1705). Comparing IgG levels between infected non-vaccinated and non-infected vaccinated individuals indicated that natural infections resulted in higher, equal or lower IgG levels than those obtained upon completion of primary vaccination with CoronaVac (*p* = 0.0001), ChAdOx1-S (*p* = 0.0603) and BNT162b2 (*p* < 0.0001), respectively.

### 3.5. IgG Response to Infection vs. Multiple Dose Vaccination

The complete dataset was used to evaluate the effect of natural infections and multiple vaccine doses on the levels of anti-spike IgG. Participants were grouped accordingly to the number of vaccine doses received with all types of vaccines combined. In the non-infected group, participants presented increased levels of IgG reactive to spike protein according to the number of vaccine doses (Figure 3B). The pairwise comparison with increasing dose number (i.e., 0 vs. 1; 1 vs. 2, etc.) were all statistically significant (*p* < 0.0001). Within the infected group, increased IgG levels were significant (*p* < 0.0001) only for the 2nd dose vs. booster (3rd dose) comparison (Figure 3B).

For pairwise comparisons, the same number of doses between infected and non-infected individuals showed that the natural infection increased IgG levels significantly (*p* < 0.0001) among unvaccinated and in those individuals that received the 1st and 2nd doses. Upon administration of the 3rd dose, IgG levels became similar (*p* > 0.99) when infected vs. non-infected groups were compared.

### 3.6. Effect of the Booster Dose (3rd Dose) According to the Vaccine Type

We evaluated the effect of the booster dose according to the type of vaccine applied. Participants who declared to have taken the 3rd dose of the vaccine in the time frame between 10 and 240 days before sampling were considered. Given that natural infections are more likely to be asymptomatic after the 2nd and 3rd vaccine doses, participants were included despite the declaration of previous infection in these analyses. For most participants (90%), the 3rd dose was BNT162b2 in either homologous or heterologous combinations. A small number of participants (8.7%) had a 3rd dose of ChAdOx1-S homologous combined with ChAdOx1-S during primary vaccination. Other combinations were present in small numbers and thus were not considered in the comparisons.

The booster heterologous vaccination regimes ChAdOx1-S + BNT162b2 or CoronaVac + BNT162b2 were very effective in augmenting the IgG levels and spike protein seroconversion rates in comparison to primary vaccination (*p* < 0.0001). In fact, the antibody levels achieved were similar to those obtained with the homologous BNT162b2 + BNT162b2 regime (Figure 4).

The 3rd dose homologous regimes ChAdOx1-S + ChAdOx1-S and BNT162b2 + BNT162b2 were able to sustain, but not increase, the IgG levels in comparison to primary vaccination (Figure 4). These data indicate that the homologous regime BNT162b2 + BNT162b2 was superior to ChAdOx1-S + ChAdOx1-S concerning levels of IgG reactive to spike protein (*p* < 0.0001) (Figure 4).

### 3.7. Principal Component Analysis

The full data set was subjected to principal component analysis (PCA) to identify the major variables. Dimension 1 (*x* axis), which explained 27% of the variation, had as main vectors the variables: vaccination, number of doses, time after doses and IgG signal to Spike and RBD (Appendix A). All these variables pointed to the positive side of the *x* axis, suggesting correlation among as expected and confirmed the trend reported in previous analyses. Dimension 2 (*y* axis) explained 11% of the variation and had as the main vectors the variables: diagnostic for COVID-19, the time after the positive diagnostic and the signal of IgG reactive to Nucleocapsid. The vectors of these two variables were nearly superimposed (Appendix A), indicating excellent correlation. The PCA confirmed that gender and city of residence resulted in negligible vectors which did not influence dataset variability, whereas age had a minor contribution.

## 4. Discussion

Global efforts for COVID-19 resulted in an extraordinary number of vaccines candidates being developed. Many countries adopted vaccines based on different technologies from different manufacturers. To understand how these different vaccines perform in a real-world scenario is an outstanding question. It is assumed that the levels of IgG reactive to spike protein and RBD can be used to reliably predict vaccine efficacy [9,10].

Previous studies investigating COVID-19 vaccine immunogenicity have focused on a particular cohort, type of vaccine and/or pre-defined dose intervals. The present work was primarily defined by a project intended to improve public health awareness, with an objective to stimulate the interest of citizens in vaccination. As such, our cohort was based on a broad range of participants, originating from a real-world scenario, where different vaccines were used for different priority groups in a context concurring with natural infections and vaccine hesitancy.

We first analyzed the evolution of SARS-CoV-2 infections and vaccination by using seroconversion to nucleocapsid and spike protein as a proxy, respectively. The data allowed us to estimate that more than a quarter of the population had been infected by SARS-CoV-2 in the first trimester of 2022 (Appendix A). The effectiveness of the vaccination program concerning spike protein immunogenicity is clear, as spike IgG seroconversion achieved >95% in February 2022, remaining at this level up to the end of this study in June 2022 (Appendix A). Despite the high spike protein seroconversion rates, a sharp increase in COVID-19 cases were detected between January and March 2022 (Appendix A). These cases were attributed to the omicron variant, which was predominant in the region at this time [20] and has the documented ability to evade neutralizing antibodies raised by vaccines and/or prior infections [21,22].

Despite the high number of COVID-19 cases between January and March 2022, it was not reflected by an increased number of deaths (Appendix A). The lower fatality rate can be partially attributed to the fact that the omicron variant causes less severe infections due to reduced viral replication in the lungs [23]. Furthermore, the presence of pre-existing spike binding antibodies in the population (Appendix A), despite the low neutralizing activity against omicron [13], is likely to be a key factor in decreasing case fatality by reducing viral replication through Fc-mediated processes [3].

The immunogenicity of the different vaccines upon completion of primary vaccination was compared in the population. This analysis revealed that the levels of IgG reactive to spike and S1 RBD were higher in those participants vaccinated with BNT162b2, followed by ChAdOx1-S and CoronaVac, in agreement with other studies [24,25].

Quite remarkably, the seroconversion rates for spike binding IgG antibodies determined in our study (Figure 1A) is in excellent agreement with vaccine efficacy numbers reported previously. Vaccine efficacies vs. seroconversion rates concerning primary vaccination were: BNT162b2 95% [4] vs. 99%; ChAdOx1-S 71% (study in Brazil) [5] vs. 81%; and Coronavac 50–55% (study in Brazil) https://www.bbc.com/news/world-latin-america-55642648 [7] (accessed on 1 May 2022) vs. 52%. This was also observed after booster doses. Combining CoronaVac + BNT162b2 resulted in vaccine efficacy against infection of 93% (study in Brazil) [7], whereas spike protein seroconversion was 94% (Figure 4). The above-mentioned data reinforces the idea that vaccine efficacy correlates with the levels of anti-spike IgG [9,10], furthermore it suggests that vaccine efficacy reported in clinical studies can be extrapolated to the real-word scenario in Brazil. It should be cautioned, however, that a positive spike protein IgG test may not represent the ability to evade symptomatic infection, especially in the light of emerging SARS-CoV-2 variants which exhibit exceptional ability to evade pre-existing antibodies.

We noted a significant negative correlation between age and interval, with the levels of IgG reactive to spike and RBD, after the 2nd dose of BNT162b2. Strikingly, this effect could not be detected in participants vaccinated with either ChAdOx1-S or CoronaVac (Figure 2 and Appendix A). It is not clear, however, if the lack of negative correlation in the case of ChAdOx1-S and CoronaVac is linked to the fact that these vaccines elicited a much lower initial response than BNT162b2. It is important to note that our study is limited by the longitudinal data being determined amongst different participants, who were predominantly younger than 60 years. Furthermore, there were a limited number of samples converting >180 days after vaccination in our study.

The data reported here confirm previous findings that natural infections act as an additional booster to raise the levels of IgG reactive to spike [13]. Such effect was clearly observed in those individuals who completed primary vaccination with CoronaVac and ChAdOx1-S, but not in the case of BNT162b2, in which primary vaccination alone already elicited a high humoral response (Figure 3).

The application of a 3rd vaccine dose raised the levels of anti-spike IgG and seroconversion rates. The levels of IgG after the 3rd vaccine dose were at such a high scale, that no additional positive effects could be observed with concurring natural infection (Figure 3). The combination of different types of vaccines during primary vaccination and booster showed that the use of BNT162b2 as a booster resulted in increased seroconversion rates and IgG levels when primary vaccination was completed with CoronaVac or ChAdOx1-S, which is in agreement with previous findings in Brazil [26]. An equivalent increase in IgG was not detected when ChAdOx1-S was used as a booster in a homologous vaccination regime (Figure 4). 

One important limitation of this study was the fact that participants were not randomly selected in the population, they spontaneously volunteered and came to the study site. Hence, populational bias may apply. Another aspect to consider is whether longitudinal comparative analysis with waning IgG levels over time in this prospective cohort is representative of the general population.

## 5. Conclusions

In conclusion, assuming the premise that the anti-spike protein IgG levels act as a proxy for vaccine efficacy and protection against COVID-19, our time series spike protein seroconversion data confirmed the effectiveness of the vaccination program in Brazil was key to control the SARS-CoV-2 pandemic. Of the vaccines used in Brazil, BNT162b2, which is based on the novel mRNA technology, performed better to elicit anti-spike protein IgG after primary vaccination and as a booster dose, and thus should be recommended as a booster whenever available.

## Figures and Tables

**Figure 1 vaccines-11-00021-f001:**
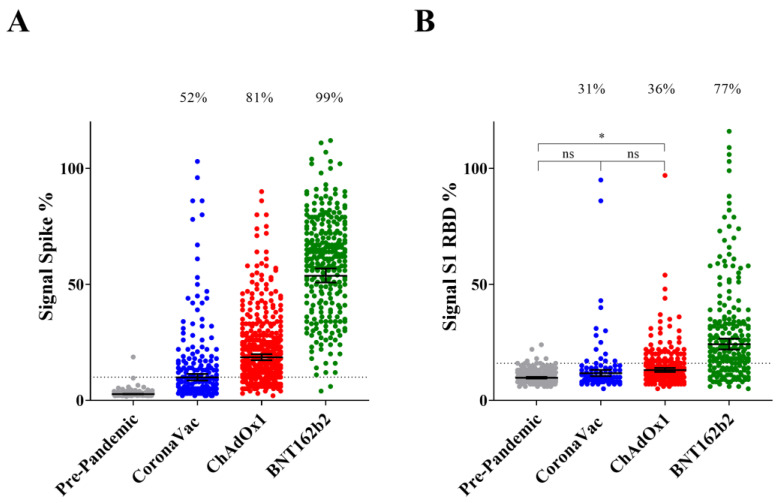
The IgG response to spike protein and S1 RBD accordingly to the vaccine type. IgG levels reactive to spike protein (**A**) and S1 RBD (**B**) in participants negative for COVID-19 who completed primary vaccination (2nd dose) between 10 and 240 days. Bars represent the geometric mean with 95% CI, the dashed line indicates the seropositive cutoff. Pre-pandemic samples were plotted as controls. One-way ANOVA analysis was performed, all comparisons were significant at *p* < 0.0001 except when indicated in the graph (* *p* < 0.05), (ns, not significant). Numbers above the graph indicate seroconversion rates. The dashed line indicates the seropositive assay cutoff.

**Figure 2 vaccines-11-00021-f002:**
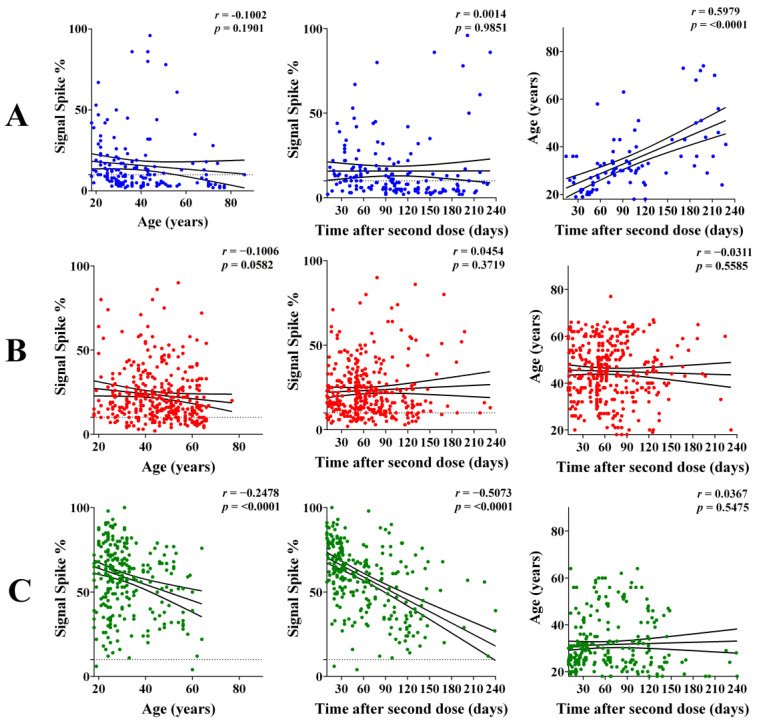
Correlation between the levels of IgG reactive to spike and the age or time after the second vaccine dose. The IgG levels were determined in participants negative for COVID-19 who completed primary vaccination (2nd dose) between 10 and 240 days. (**A**) Correlations between IgG levels and age. (**B**) Correlations between IgG levels and time after the second dose. (**C**) Correlations between age and time after the second dose. The linear regression with 95% CI and Pearson’s correlation coefficient is indicated. Color code, green BNT162b2, red ChAdOx1-S, blue CoronaVac.

**Figure 3 vaccines-11-00021-f003:**
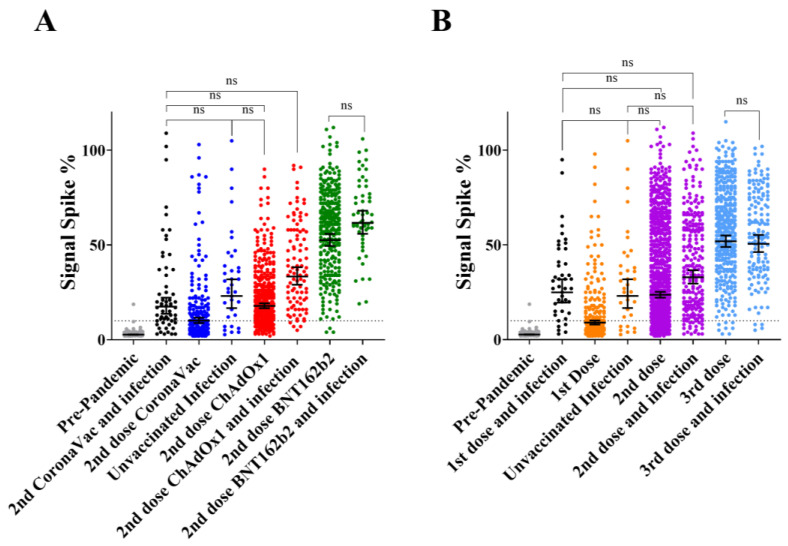
The levels of IgG reactive to spike protein in response to infection and vaccination. (**A**) Participants that completed the primary vaccination (2nd dose) were grouped accordingly to the vaccine type. Those participants who self-declared a previous positive COVID-19 test are indicated as infection cases. (**B**) Participants were grouped accordingly to the number of vaccine doses. The dashed line indicates the seropositive assay cutoff. Pre-pandemic samples were plotted as naïve controls. Bars represent the geometric mean and 95% CI. One-way ANOVA analysis was performed, and comparisons were all significant (*p* < 0.02) unless indicated as not significant (ns).

**Figure 4 vaccines-11-00021-f004:**
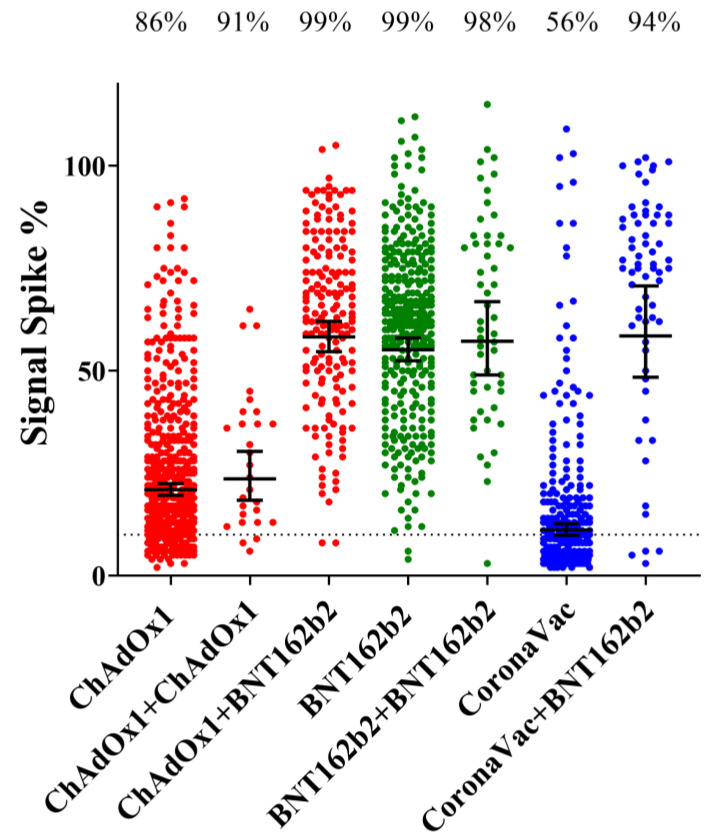
IgG levels reactive spike after completion of primary vaccination and after the booster dose accordingly to the vaccine type. IgG levels reactive for spike in all participants (including those negative and positive for natural infections) who completed primary vaccination (2nd dose) between 10 and 240 days are indicated accordingly to the vaccine type. The IgG levels in participants who took a 3rd booster dose is indicated with + followed by the vaccine type of the booster. Only participants who took the booster dose between 10 and 240 days were considered. The dashed line indicates the seropositive assay cutoff. The geometric mean and 95% CI are represented by the bars. Number indicates the seroconversion rates.

## Data Availability

The data supporting the findings of this study are available on request from the corresponding author (L.F.H.).

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
