# Peer review of "Population-Based Analysis of the Immunoglobulin G Response to Different COVID-19 Vaccines in Brazil"

_vaccines, 2022, doi:10.3390/vaccines11010021_

Round 1
Reviewer 1 Report
1. Please remove/combined “This study was set up to determine the humoral immunoglobulin G response to different SARS-CoV-2 vaccines available in Brazil in a real-world scenario. Weekly sampling campaigns were open to the public between January 2021 to June 2022 in the city of Matinhos, in the coastal region of Paraná state, in the south of Brazil” in the results with “2.1 Study design and sampling.”
2. In line 166 and other places, the seroconversion definition is unclear.
3. In line 161, the authors mentioned “Only participants who declared to have taken the second dose of the vaccine in the time frame between 10 and 240 days before sampling were considered.” As we know that the IgG level grows rapidly soon after vaccination but the level of IgG also drops drastically after the vaccination, so it is important to take the period after 2nd vaccination for the analysis. This is amplified in figure 1 where a number of individuals are showing low level of IgG in each of the different vaccines, would these “low-level IgG” be low because the time since their 2nd vaccination is long or because these individuals do not respond to the vaccine?
Author Response
Reviewer 1
Please remove/combined “This study was set up to determine the humoral immunoglobulin G response to different SARS-CoV-2 vaccines available in Brazil in a real-world scenario. Weekly sampling campaigns were open to the public between January 2021 to June 2022 in the city of Matinhos, in the coastal region of Paraná state, in the south of Brazil” in the results with “2.1 Study design and sampling.”
We have removed/combined part of this text
In line 166 and other places, the seroconversion definition is unclear.
We have added the definition of seroconversion in the Material and Methods section.
- In line 161, the authors mentioned “Only participants who declared to have taken the second dose of the vaccine in the time frame between 10 and 240 days before sampling were considered.” As we know that the IgG level grows rapidly soon after vaccination but the level of IgG also drops drastically after the vaccination, so it is important to take the period after 2nd vaccination for the analysis. This is amplified in figure 1 where a number of individuals are showing low level of IgG in each of the different vaccines, would these “low-level IgG” be low because the time since their 2nd vaccination is long or because these individuals do not respond to the vaccine?
We completely agree with the referee that the time point after vaccination is important for immunity. Indeed, we have depicted this important information in Fig.2 which shows the IgG response accordingly to the time after the second dose for each type of vaccine available in Brazil. In, the Fig. S5, the same information depicted in Fig.1 is presented with a short time interval after the second dose. As can be seen in this figure the response is similar to Fig.1, hence we can assume that those individuals with low IgG level in Fig.1 are those that did not respond to the vaccination.
Reviewer 2
Suggest adding the serologic instruments and reagents in 2.2.
The serologic assessment is a principal technique used to investigate the major data. You cannot only cite previous publications without mentioning them. For venipuncture, transfer and storage are acceptable to cite previous publications without mention.
We have added more information describing the serological technique used.
Suggest considering the significant figures. Some of the p-value results are>0.001 (3 decimals) but another is 0.54 (2 decimals). Suggest us the same decimals throughout the manuscript to make it consistent.
We have changed the figures and now we use the same number of decimals in all figures.
3 Figure S4. What is the p = 0.00?. It is impossible the p-value is equal to 0. Please revise again.
We have corrected the p value.
Comments.
Suggest using a common form of "Pfizer-BioNTech" instead of "Pfizer/BioNTech".
Done.
ChAdOx1 is the adenoviral vector platform. ChAdOx1-S is the viral vector encoded by the target gene of the SARS-CoV-2 spike protein. Suggest using "ChAdOx1-S" instead of "ChAdOx1".
Done.
Suggest using "AstraZeneca-University of Oxford" instead of "Oxford/AstraZeneca".
Done.
Suggest using "BNT162b2" instead of "BNT16b2" .
Done.

Reviewer 2 Report
This manuscript is a good methodology and is well-structured. The results are informative, visualisation is clear and attractive. Overall am satisfied with this content.
Major concerns.
1. Suggest adding the serologic instruments and reagents in 2.2.
The serologic assessment is a principal technique used to investigate the major data. You cannot only cite previous publications without mentioning them. For venipuncture, transfer and storage are acceptable to cite previous publications without mention.
2. Suggest considering the significant figures. Some of the p-value results are>0.001 (3 decimals) but another is 0.54 (2 decimals).
Suggest us the same decimals throughout the manuscript to make it consistent.
3 Figure S4. What is the p = 0.00?
It is impossible the p-value is equal to 0. Please revise again.
Comments.
1. Suggest using a common form of "Pfizer-BioNTech" instead of "Pfizer/BioNTech".
2. ChAdOx1 is the adenoviral vector platform. ChAdOx1-S is the viral vector encoded by the target gene of the SARS-CoV-2 spike protein.
Suggest using "ChAdOx1-S" instead of "ChAdOx1".
3. Suggest using "AstraZeneca-University of Oxford" instead of "Oxford/AstraZeneca".
4. Suggest using "BNT162b2" instead of "BNT16b2" .
Author Response

(The authors gave the same response as above.)
